# Leptospirosis Risk Assessment in Rodent Populations and Environmental Reservoirs in Humanitarian Aid Settings in Thailand

**DOI:** 10.3390/microorganisms13010029

**Published:** 2024-12-27

**Authors:** Panadda Krairojananan, Kasima Wasuworawong, Surachai Leepitakrat, Taweesak Monkanna, Elizabeth W. Wanja, Silas A. Davidson, Betty K. Poole-Smith, Patrick W. McCardle, Alyssa Mann, Erica J. Lindroth

**Affiliations:** 1Department of Entomology, Walter Reed Army Institute of Research-Armed Forces Research Institute of Medical Sciences, Bangkok 10400, Thailand; kasimaw.ca@afrims.org (K.W.); surachail.fsn@afrims.org (S.L.); taweesakm.ca@afrims.org (T.M.); alyssa.mann.mil@afrims.org (A.M.); erica.lindroth.mil@afrims.org (E.J.L.); 2Entomology Branch, Center for Infectious Disease Research, Walter Reed Army Institute of Research, Silver Spring, MD 20910, USA; wanjaeli@yahoo.com; 318th MEDCOM, Fort Shafter, Honolulu, HI 96858, USA; silas.a.davidson.mil@army.mil; 4Army Medical Department Student Detachment, San Antonio, TX 78288, USA; poolesmith@rand.org; 5Defense Centers for Public Health-Aberdeen, DHA Public Health, Aberdeen Proving Ground, Aberdeen, MD 21010, USA; patrick.w.mccardle.mil@health.mil

**Keywords:** leptospirosis, rodent populations, environmental reservoirs, one health, surveillance program, risk assessment, Thailand

## Abstract

Leptospirosis, a global zoonotic disease caused by *Leptospira* spp., presents high morbidity and mortality risks, especially in tropical regions like Thailand. Military personnel deployed in endemic areas, such as during the Cobra Gold Joint exercise, face heightened exposure. This study assessed *Leptospira*’s prevalence in rodents and environmental reservoirs at military training sites from 2017 to 2022. A surveillance program was conducted at Engineering Civil Assistance Program (ENCAP) training sites using real-time PCR, dark-field microscopy, and 16S rRNA gene sequencing to detect *Leptospira* in rodents and environmental samples. Results showed a 1.3% infection rate in rodents (15 of 1161), while *Leptospira* was detected in 10.2% of water samples (42 of 413) and 23.1% of soil samples (30 of 130). Diverse *Leptospira interrogans* strains circulated among rodents, and three groups of naturally circulating *Leptospira* strains were detected in environmental reservoirs. These findings underscore *Leptospira*’s survival and transmission potential within exercise sites, informing Force Health Protection (FHP) decisions. By integrating pre-exercise data on primary hosts and environmental reservoirs with historical local outbreak records and research on risk factors, this study identifies key areas for public health intervention and potential mitigation strategies.

## 1. Introduction

Leptospirosis is a zoonotic disease with significant global impacts causing approximately one million cases and 58,900 deaths annually, with a case fatality rate of 6.85% [1]. Clinical symptoms range from non-specific fever to organ failure and death, so diagnostics are required to confirm cases of the disease, and leptospirosis is underreported, especially in endemic areas [2].

The disease thrives in tropical climates where warm temperatures, high rainfall, and flooding facilitate the bacteria’s life cycle [3]. *Leptospira* species are categorized into pathogens, intermediates, and saprophytes based on their virulence status [4], and they persist in animal kidneys and excreted urine, contaminating the soil and water [5]. Transmission to humans occurs through contact with infected animals or contaminated water sources, with occupational exposure in agriculture and recreational activities being the main risk factors [6].

In Thailand, leptospirosis is often linked to seasonal flooding, which significantly affects transmission rates [7]. Most cases involve agricultural workers exposed to reservoir animals and contaminated environments [8,9,10]. The annual notification rate from 2010 to 2020 was 1.41 cases per 100,000 population with a range of 1.87 to 7.76 cases. Although the Thai Bureau of Epidemiology, the Department of Disease Control, and the Ministry of Public Health (MoPH) [11] mandate leptospirosis reporting, only hospital cases are reported, so the less severe and self-limiting cases are excluded. Consequently, human leptospirosis cases are underreported relative to the true national or regional burden of disease.

Military personnel are at risk for contracting leptospirosis through contact with the soil and water, especially during exercises in endemic areas like Thailand. Despite the fact that a pre-exposure prophylaxis program could decrease infections [12,13], infection still occurs, often leading to misdiagnosis and complications [14,15,16,17]. Because leptospirosis is a treatable and preventable disease, a prophylactic regimen has the potential to reduce the risk of medical complications, particularly for expeditionary military personnel [18]. Therefore, we implemented a pre-exercise surveillance program to monitor the prevalence of leptospirosis in training areas in Thailand to quantify exposure, identify high-risk areas for avoidance, and ultimately reduce the incidence of cases in military and local civilian populations.

Cobra Gold (CBG) is an annual multinational military exercise co-hosted by the Royal Thai Armed Forces and the United States Indo-Pacific Command. It presents leptospirosis risks due to environmental exposure during training activities. This study aimed to evaluate the prevalence of *Leptospira* in rodent populations and environmental reservoirs within the zones where the Engineering Civic Assistance Program (ENCAP) team participated in the CBG exercise as part of continued civic action and humanitarian assistance efforts. We collected samples from rodents, water bodies, and the soil in agriculture fields and marshes to assess *Leptospira*’s circulation and the associated risk in these training areas.

## 2. Materials and Methods

### 2.1. Ethics Statement

This study was approved by the Institutional Animal Care and Use Committee (IACUC) at the Armed Forces Research Institute of Medical Science (AFRIMS) in Bangkok, Thailand, under protocol numbers 15-06 (approval date: 15 June 2015), 18-04 (approval date: 3 August 2018), and 21-10 (approval date: 25 August 2021). 

### 2.2. Sampling Sites

Surveillance was conducted annually before the joint military training exercise Cobra Gold in Thailand. Four to five provinces were selected each year, with samples collected from one to two locations per province. Details regarding the survey period, the rodents captured, and the infestation rate are provided in Appendix A. The locations of training sites (2017–2022) and reported passive leptospirosis cases from the Thai MoPH are shown, with each year’s surveillance sites distinguished by color (Figure 1).

### 2.3. Small Mammal Trapping

One hundred wire live traps were set nightly for three consecutive nights at each study site both indoors and outdoors. Verbal consent was obtained from landowners for traps placed on private property. Traps were baited with dried fish, fruit, and other food items attractive to local rodent species, placed 5 m apart, and tracked with a Garmin GPS tracker. Traps were set by 6 p.m. and checked by 7 a.m. to minimize rodent stress. Trap success (TS) was calculated as TS (%) = Tc/Tn × 100, where Tc is the number of trapped rodents and Tn is the number of trap nights [19].

### 2.4. Rodent Identification and Specimen Processing

Rodents were euthanized using carbon dioxide, identified to the species level [20,21,22], and dissected to collect blood, kidney tissue, and urine. Kidneys were used for *Leptospira* culture and DNA extraction. Urine was collected aseptically for culture.

### 2.5. Leptospira Culture and Isolation

Urine (if available) and kidney tissue were cultured in 5-fluorouracil-containing Ellinghausen-McCullough-Johnson-Harris (EMJH) medium (Difco, NJ, USA) with 5% rabbit serum (Thermo Fisher Scientific, MA, USA). Samples were incubated at 30 °C for 16 weeks and examined bi-weekly under a dark-field microscope (ZEISS Axio Lab.A1, Carl Zeiss Microscopy GmbH, Jena, Germany). Viable *Leptospira*, identified by their spiral shape and motility, were isolated on solid EMJH medium and prepared for further analysis.

### 2.6. Environmental Sampling

Water (30 mL) and soil (10 g) samples were collected from training sites near rodent traps. Soil sampling began in 2020. Samples were incubated in EMJH medium at 30 °C and checked daily for *Leptospira* growth, followed by filtering and single-colony isolation. The environmental parameters, the pH, and the temperature were measured in the environmental samples using the HI 99121N Direct Soil and Water Portable pH/Temperature Meter (Hanna Instruments Inc., Smithfield, RI, USA).

### 2.7. DNA Extraction

Different Qiagen kits were used for genomic DNA extraction to ensure optimal recovery from each sample type. Rodent kidney tissues were processed using the QIAamp DNA Mini Kit (Qiagen, Valencia, CA, USA) following the manufacturer’s instructions. For water samples, 30 mL of water was centrifuged at 3000× *g* for 30 min, and the resulting pellet was resuspended in 140 µL of sterile water. DNA from these samples was extracted using the QIAamp Viral RNA Mini Kit (Qiagen, Valencia, CA, USA), adhering to the protocol for urine samples. Soil samples (0.25 g) were processed using the Qiagen PowerSoil DNA Extraction Kit (Qiagen, Valencia, CA, USA), as per the manufacturer’s instructions. For *Leptospira* isolates, DNA was extracted with the DNeasy UltraClean Microbial Kit (Qiagen, Valencia, CA, USA), following the protocol designed for Gram-negative bacteria. In each extraction batch, ultrapure water was used as a blank control to monitor for any cross-contamination. Extracted DNA samples were stored at −20 °C until they were used in quantitative PCR assays.

### 2.8. Leptospira DNA Detection and Species Identification

Two TaqMan assays were used: one targeting the lipL32 lipopolysaccharide (LPS) [23] and another targeting *Leptospira* 16S rRNA (*rrs*) [24]. Briefly, each 25 µL reaction mixture contained 12.5 µL of 2× TaqMan™ Multiplex Master Mix (Thermo Fisher Scientific, Waltham, MA, USA) and 5 μL of DNA template. For detecting *Leptospira* DNA in rodents, the LPS real-time assay included 0.1 µM of LPS-probe, 0.2 µM each of LPS-forward and LPS-reverse primers, and an internal control set of the beta actin (BA) gene: 0.025 µM of BA-probe and 0.05 µM each of BA-forward and BA-reverse primers in the master reaction. To assess *Leptospira*’s persistence in environmental samples, the *rrs* assay included 0.1 µM of 16S probe and 0.25 µM each of Lepto F and Lepto R primers. qPCR inhibition was monitored using qPCR Extraction Control (Bioline, London, UK) as an internal amplification control following the manufacturer’s instructions. Positive controls included DNA from reference *L.*
*interrogans* culture and *Leptospira*-infected rodent kidney tissues from prior surveillance studies, while negative controls included DNA from non-pathogenic *L.*
*biflexa* culture and non-infected rodent tissue. All samples were run in duplicate with non-template controls to detect any contaminating DNA. Real-time PCR was performed using the Chromo4™ System (Bio-Rad, Hercules, CA, USA) with cycling conditions of 95 °C for 1 min followed by 45 cycles at 95 °C for 15 s and 60 °C for 30 s.

Positive *lipL32* and *rrs* qPCR assay samples were confirmed using a nested single-tube PCR assay targeting the partial 16S rDNA gene, as previously described [25]. *L.*
*interrogans* genomic DNA served as a positive control, and DNase-free water was used as a negative control in conventional PCR. Amplicons were analyzed using 1.5% gel electrophoresis stained with GelStar (Lonza, Walkersville, MD, USA). PCR products were purified using the DNA Clean & Concentrator-5 (Zymo Research, Irvine, CA, USA) and sent to Macrogen Inc. (Seoul, Republic of Korea) for sequencing. DNA sequences of partial *rrs* amplicon were analyzed using Sequencher^TM^ v5.0 software (Gene Codes Corporation, Ann Arbor, MI, USA), trimmed to the 433 base pair length fragments. For molecular identification of *Leptospira* species, sequences were compared to available *Leptospira* sequences in the GenBank database using the BLASTn algorithm. Partial 16S rRNA gene sequences from *Leptospira* obtained in this study were deposited in GenBank, with the accession numbers listed in Appendix A. Maximum likelihood trees were constructed using the Kimura 2-parameter nucleotide substitution model for partial 16S rRNA gene sequences. Bootstrap analysis with 1000 replicates was performed to assess node support in MEGA v.11 [26]. Reference sequences retrieved from GenBank were represented in the phylogenetic tree as *Leptospira spp._Accession number*, while the 16S rRNA sequences from this study were represented as *collection number_Host species* (for rodent)_*collecting province* or *collection number* (for water or soil)_*collecting province*.

### 2.9. Molecular Characterization of Recovered Leptospira Strains

Species identification of *Leptospira* isolates, including 4 rodent isolates, 27 water isolates, and 40 soil isolates, was performed by amplifying bacterial 16S rDNA fragments. PCR amplification was conducted using 300 nM 16S rRNA primers fD1/rD1 [27]. The amplification process was carried out on a SimpliAmp™ Thermal Cycler (Applied Biosystems, Foster City, CA, USA) with an initial denaturation at 94 °C for 2 min, followed by 35 cycles of 92 °C for 15 s, 63 °C for 30 s, and 68 °C for 2 min, with a final extension at 68 °C for 5 min. Amplicons were analyzed using 1.5% gel electrophoresis, purified, and sent to Macrogen Inc. (Seoul, Republic of Korea) for sequencing with the primers lepto16S11f, lepto16S505f, and lepto16S1006f, as previously described [28]. Between 90% and 93% of the 16S rRNA gene sequences from *Leptospira* isolates were deposited in GenBank (accession numbers provided in Appendix A). Maximum likelihood trees were constructed using the Kimura 2-parameter model based on the 16S rRNA gene sequences, with bootstrap analysis performed using MEGA v.11 [26]. Reference sequences retrieved from GenBank were represented in the phylogenetic tree as *Leptospira spp._Accession number*, while the isolates from this study were represented as *collection number_Host species (for rodent)_collecting province* or *collection number (for water or soil)_collecting province*.

### 2.10. MLST Typing of Pathogenic Leptospira

MLST typing was performed on pathogenic *Leptospira* isolates using Scheme 1 [29]. To directly type pathogenic *Leptospira* from positive kidney samples, we applied the previously published MLST with nested PCR method for Scheme 1, using the primer sets, annealing temperatures, and cycling conditions as described by Weiss et al. [30].

### 2.11. Statistical Analysis

Statistical analyses were performed using IBM SPSS Statistics 26.0 (IBM Corp., New York, NY, USA). The mean trapping success in rodent populations and its 95% confidence interval (CI) were calculated. Fisher’s exact test was used to assess the association between *Leptospira* prevalence and rodent species. A *p*-value of less than 0.05 was considered statistically significant.

## 3. Results

### 3.1. Location and Passive Surveillance Data of Pre-Surveillance Sites

Thailand’s tropical climate is characterized by distinct wet (June–October) and dry (November–May) seasons. Leptospirosis is a notifiable disease monitored by the Thai MoPH. Between 2016 and 2021, the annual incidence of leptospirosis ranged from 1.65 to 5.31 cases per 100,000 people, with the highest reports from the northeast and the south, driven by environmental factors and agricultural activities, particularly rice cultivation [11]. The study was conducted during the dry season (November–January) before the CBG training exercise. Trapping occurred in 31 sites across 15 provinces where military humanitarian activities were held from 2017 to 2022 (Appendix A and Figure 1). These rural locations were mostly sites of school construction. Based on MoPH passive surveillance data, most provinces involved in CBG training had lower leptospirosis incidence than the national average, with the exception of Chantaburi and Krabi (Figure 2).

### 3.2. Diversity and Prevalence of Leptospira in Trapped Small Mammal Populations

#### 3.2.1. Abundance of Small Mammals

Over six consecutive years, 1161 small mammals were trapped across 31 sites in 15 provinces (Appendix A). These mammals were captured near activity sites to assess leptospirosis’s prevalence in these primary reservoirs. Trapping was conducted with the number of trap nights per site ranging from two to four and yearly trap nights ranging from 8 to 21. This resulted in a trapping success rate of 12.1%, 95% CI (10.2, 14.0) per year across all sites (Appendix A). The relative abundance of small mammals varied across surveillance sessions, with the highest abundance observed in the pre-surveillance session for CBG21. Excluding CBG17, the 95% confidence interval for relative abundance among the remaining five surveillance sessions ranged from 12.4 to 15.7, with an average of 13.7.

Fourteen species (Appendix A) were captured, including twelve species from the order Rodentia and two from the order Scandentia. *Rattus rattus* (n = 597, 51.4%) was the most common species (Figure 3), followed by *Bandicota savilei* (n = 192, 16.5%) and *Bandicota indica* (n = 179, 15.4%). With the exception of the site in Chachoengsao, *R.*
*rattus* was found at nearly all of the sites. Furthermore, these three species were consistently trapped each year, while *Berylmys bowersi*, *Rattus norvegicus*, *Mus cervicolor*, *Niviventer fulvescens*, *Mus caroli*, and *Tupaia glis* were captured less frequently.

Chantaburi and Rayong were consistently selected as ENCAP project focal areas. Over six years, the average trapping success rate was 9.4%, 95% CI (4.7, 14.1) in Chantaburi and 12.6%, 95% CI (3.1, 22.2) in Rayong. Twelve small mammal species were identified, with *R.*
*rattus* being the most abundant throughout the surveillance period. Seven species, including *B.*
*indica*, *B.*
*savilei*, *B.*
*berdmorei*, *M.*
*berdmorei*, *N.*
*fulvescens*, *R.*
*exulans*, and *R.*
*rattus*, were trapped in both provinces. In Rayong, additional species, such as *B.*
*bowersi*, *Maxomys surifer*, *M.*
*caroli*, and *Tupaia belangeri,* were caught, while *R.*
*norvegicus* was only trapped in Chantaburi (Appendix A).

#### 3.2.2. Molecular Detection and Identification of *Leptospira* spp.

Pathogenic *Leptospira* spp. were detected in 1.3% (15/1161) of the trapped rodents, with the following host reservoirs identified: *R.*
*rattus* (n = 7, 0.6%), *B.*
*indica* (n = 3, 0.3%), *B.*
*savilei* (n = 2, 0.2%), *B.*
*berdmorei* (n = 2, 0.2%), and *M.*
*cervicolor* (n = 1, 0.1%) (Appendix A). Fisher’s 2 × 2 exact test indicated no significant association between *Leptospira* prevalence and the specific host species (*p* = 0.0538 to 1.000).

*Leptospira* carriage was absent in small mammals captured prior to CBG17 but present in CBG18-22 (Figure 4). The geographical distribution of the infected rodents is detailed in Appendix A: Chantaburi (n = 5, 2.6%), Krabi (n = 5, 7.6%), Nakhon Ratchasima (n = 1, 5.6%), Rayong (n = 1, 0.3%), Tak (n = 2, 2.4%), and Trat (n = 1, 0.7%). *Leptospira* prevalence in rodents included *R.*
*rattus*, *B.*
*berdmorei*, and *B.*
*savilei* trapped from Chantaburi; *R.*
*rattus* and *B.*
*indica* from Krabi; *B.*
*indica* and *B.*
*savilei* from Tak; *M.*
*cervicolor* from Nakhon Ratchasima; and *R.*
*rattus* from Rayong and Trat.

Molecular typing of the 15 *Leptospira*-positive samples identified two human pathogenic *Leptospira* species, *L.*
*interrogans* (n = 13) and *L.*
*borgpetersenii* (n = 2), based on comparisons with GenBank sequences using the Blast algorithm and phylogenetic analysis (Figure 5). Nested MLST analysis amplified all seven loci for 11 of the 15 samples (Appendix A). In Chantaburi, more sequence types (STs) were found compared to Krabi, with ST205 as the predominant type. New STs, ST341 and ST342, were identified in *L.*
*interrogans*-infected *B.*
*berdmorei*. The predominant clone, ST34, associated with a northeastern Thailand outbreak in 2000, was found in qPCR-positive *B.*
*savilei* from Tak and Chantaburi.

#### 3.2.3. Identification of Circulating *Leptospira* spp. from the Trapped Rodents

Kidney and urine samples from trapped small mammals were cultured in EMJH cultures with 5-FU, resulting in four successful *Leptospira* spp. Isolations, with one from a kidney and three from urine. These isolates were molecularly typed using nearly the full-length 16S rRNA gene. Phylogenetic analysis (Figure 6) revealed that all four isolates were most similar to *L. interrogans*. These *Leptospira* isolates were geographically distributed as follows: one from *B. indica* in Chantaburi, one from *R. rattus* in Rayong, and two from *B. indica* and *R. rattus* in Trat. MLST analysis identified four distinct circulating *L.*
*interrogans* sequence types (STs), ST49, ST324, ST337, and ST338, with the latter three being newly discovered in Thailand (Appendix A).

### 3.3. Leptospira in Environmental Reservoirs (Water and Soil) Around Training Sites

#### 3.3.1. Collecting Environmental Reservoirs

We tested *Leptospira* persistence in 413 water samples and 130 soil samples from exposed environmental areas. Soil samples were collected only in 2021 and 2022. The pH and temperature ranges measured in the environmental samples were 6.2–7.8 and 21.7–24.6 °C for water samples and 4.5–7.6 and 22.0–25.0 °C for soil samples.

#### 3.3.2. Molecular Detection of *Leptospira* in Water and Soil Reservoirs

*Leptospira* DNA was detected using *rrs*-qPCR in 10.2% (42/413) of water samples and 23.1% (30/130) of soil samples (Appendix A). The highest detection rate in water occurred during CBGFY18 (23.8%), with positive samples primarily from the Rayong (45%, 9/20), Lopburi (29%, 6/21), and Chantaburi (25%, 5/20) exercise sites. No *Leptospira* DNA was detected at CBGFY17 sites (Figure 7 and Appendix A). In CBGFY20, we detected *Leptospira* from exercise sites in Chantaburi (30%, 3/10) and Rayong (27%, 3/11). In CBGFY19, the highest detection rates were in Nakhonsawan (20%, 4/20) and Chantaburi (15.2%, 2/13). The *Leptospira* detection rate decreased in CBGFY21 (1.6%) and CBGFY22 (4%). Soil samples in CBGFY21 showed *Leptospira* DNA in 21.3% (13/61), and in CBGFY22, 24.6% (17/69) of samples tested positive (Figure 7 and Appendix A). *Leptospira*-positive soil samples were found in Chantaburi (25%, 2/8) and Rayong (30%, 11/37) exercise sites in CBGFY21, while in CBGFY22, *Leptospira* DNA was found at the following sites: Chantaburi (43%, 6/14), Krabi (28.6%, 4/14), Rayong (21.4%, 3/14), Saraburi (10%, 1/10), and Trat (17.6%, 3/17). To confirm qPCR results, partial sequencing of the 16S rRNA gene was successfully performed on 31 of the 42 qPCR-positive water samples. Phylogenetic analysis (Figure 8) revealed clustering into pathogenic (n = 9) and intermediate (n = 22) groups. In the pathogenic group, one water sample (WSCB102 CTB) clustered with pathogenic *L.*
*interrogens*, while eight clustered with *L.*
*yasudae*. Most water samples (67.7%, 21/31) fell into the intermediate group, with six related to *L. licerasiae*, fifteen to *L. wolffii*, and one clustered with *L. perolatii*. For soil samples, 16S rRNA gene sequencing was successfully performed on 21 of the 27 qPCR-positive samples, and 85.7% (18/21) of these samples clustered in the intermediate group, while 14.3% (3/21) clustered with *L.*
*yasudae* in the pathogenic group (Figure 9).

#### 3.3.3. Circulating *Leptospira* spp. in the Environmental Reservoirs

*Leptospira* isolates were recovered annually from water samples during surveillance, with additional isolates found in soil samples throughout the study period (Appendix A). These soil isolates were geographically distributed across all exercise sites, showing a high recovery percentage, while water isolates were recovered from only some of the exercise sites. A total of 67 (12.3%) *Leptospira* isolates were recovered from 543 soil and water samples. Identification was achieved through a combination of direct microscopic observation of characteristic helical morphology, amplification of *Leptospira*-specific 16S rRNA gene sequences via PCR, and subsequent sequence analysis.

This study investigated the circulating and phylogenetic distribution of *Leptospira* in water samples collected across surveillance sites over six years. Of the 413 samples analyzed, 27 (6.5%) yielded *Leptospira* spp. isolates through culturing, with an average recovery rate of 6.5 ± 1.4 isolates per year. Molecular phylogenetic analysis based on the 16S rRNA gene (Figure 10) revealed that most isolates (n = 17) belonged to the saprophytic group, followed by the intermediate group (n = 6) and a low virulence subclade within the pathogen group (n = 4). Further analysis identified isolates genetically related to *L.*
*kmetyi* (n = 1) and *L.*
*yasudae* (n = 3) within the pathogenic group and *L.*
*licerasiae* (n = 2), *L.*
*wolffii* (n = 2), *L.*
*hartskeerlii* (n = 1), and *L.*
*koniambonensis* (n = 1) in the intermediate group. The saprophytic group included isolates related to *L.*
*mtsangambouensis* (n = 8), *L.*
*montravelensis* (n = 2), and *L.*
*kemamanensis* (n = 6). One saprophytic isolate (WSCB25 CYP) remained unidentified, potentially representing a novel *Leptospira* species.

Similarly, 130 soil samples were collected over two consecutive years, yielding 40 (30.8%) *Leptospira* spp. isolates. Phylogenetic analyses (Figure 11) revealed clustering within the saprophytic *Leptospira* group (n = 25), the intermediate group (n = 10), and a low virulence subclade within the pathogen group (n = 5). The pathogenic isolates were genetically related to *L.*
*kmetyi* (n = 1) and *L.*
*yasudae* (n = 4). In the intermediate group, isolates were related to *L.*
*licerasiae* (n = 1), *L.*
*wolffii* (n = 2), *L.*
*hartskeerlii* (n =3), *L.*
*dzoumogneensis* (n = 1), and *L.*
*johnsonii* (n = 1). Two intermediate group isolates (CBS022CTB and CBS069SRB) showed no clear relationship to the reference species, suggesting potentially novel species. In the saprophytic *Leptospira* group, isolates were genetically related to *L.*
*mtsangambouensis* (n = 23), *L.*
*idonii* (n = 1), and *L.*
*kemamanensis* (n = 1).

## 4. Discussion

This study confirmed the prevalence and circulation of *Leptospira* in both animal hosts and environmental reservoirs. Overall, 1.3% (15 out of 1161) of rodents were infected, predominantly with *L. interrogans* (87%) and the remainder with *L. borgpetersenii* (13%), both known human pathogens. The dominance of *L. interrogans* may correlate with higher leptospirosis cases in endemic areas in Thailand [31,32].

Across the exercise sites, the highest prevalence of *Leptospira* spp. was found in *R. rattus* (0.6%), followed by *B. indica* (0.3%) and *B. savilei* (0.2%), suggesting that *R. rattus* poses a notable risk to humans. However, when comparing infection rates by species, *B. indica* had a slightly higher infection rate (1.7%) than *R. rattus* (1.2%). Interestingly, despite being the fourth most abundant species captured*, R. exulans* showed no *Leptospira* infection in this study, contrasting with previous findings in high-incidence regions where it had a higher infection rate than *R. rattus* [33]. Environmental factors, such as geographic differences, may influence the prevalence of diseases in human populations due to variations in wildlife reservoir hosts. Because the study focused on trapping rodents near training areas and local villages, the close association of *R. rattus* with humans [34], along with its high *Leptospira* prevalence, suggests indirect exposure through urine-contaminated environment as a key factor in infection. This is particularly concerning in high-risk settings, such as areas with livestock farming, where *Leptospira* exposure risk is known to be higher [35]. In the training areas studied, livestock kept in the backyards of local residents are commonly found nearby, potentially increasing the risk of environmental contamination and human exposure.

Although the infection rate in rodents was relatively low (1.3% in this study and 3.6% in a previous one), many mammals can serve as reservoirs for *Leptospira*, shedding the bacteria into the environment and contributing to contamination of the soil and water sources. Understanding the broader factors that influence *Leptospira*’s prevalence beyond specific rodent infections is crucial. Variations in animal host populations, along with *Leptospira*’s ability to adapt to different hosts across geographies, can significantly impact human disease prevalence via environmental contamination. Our dry-season survey (November–January), conducted before the CBG training exercise, likely avoided the effects of flood-related transmission [7]. However, sampling only during the dry season may underestimate *Leptospira*’s prevalence, as transmission rates typically rise during the rainy season. Additionally, most training sites were not located in leptospirosis-endemic areas, which may explain the lower prevalence compared to our previous 3.6% report in northeastern Thailand, where *B.*
*indica* had the highest carrier rate [33]. Differences in rodent distribution and habitat may also influence *Leptospira*’s prevalence accordingly [36].

Understanding the spatial distribution of pathogenic *Leptospira* is crucial for implementing targeted preventive measures. Culturing fastidious *Leptospira* is time-consuming and labor-intensive; however, using the modified MLST method [30] applied directly to reservoir tissues, especially the kidney, could accelerate the identification of circulating *Leptospira* spp. in training areas. In this study, the MLST data [37] were used to compare *Leptospira* found at exercise sites with publicly available STs of epidemiologically significant strains causing human leptospirosis in Thailand. Based on these data, the distribution of *Leptospira* across the exercise areas included 10 STs, identified from both rodent culture and directly modified MLST testing. Three of the new STs (ST324, ST337, and ST338) of *L. interrogans*, although currently identified only in Thailand, may pose a transmission risk in areas where they were recovered from the rodent urine. The colonized *L. interrogans* ST49 isolate, recovered from the kidney, has also been found in Sri Lanka. According to MLST data, four STs (ST259, ST337, ST341, and ST342) are endemic strains in Thailand, while ST34 has also been found in Sri Lanka and Lao and ST205 and ST235 in Malaysia [37]. Moreover, *L. interrogans* ST34 has remained the most common cause of human leptospirosis since a major outbreak in 2000 [31]. Our surveillance also found that ST34 is a predominant strain in many rodent reservoirs, and it is linked to human leptospirosis cases in endemic areas [31]. Unfortunately, the incomplete molecular typing of all positive samples may limit our understanding of the full genetic diversity of circulating *Leptospira* strains.

Although our surveys prior to ENCAP missions take place during the dry season, some *Leptospira* spp. can survive in the environment for up to one year [38]. Four *L. interrogans* isolates from rodent populations in this study, including three recovered directly from urine, demonstrated a potential risk for human exposure through habitat contamination around training sites, even at low rodent infection prevalence. Additionally, three of the four circulating isolate types were new STs isolated in Thailand, indicating a complex epidemiology of natural *Leptospira* circulation. The environment around the training sites facilitates the transmission of *Leptospira* from the primary reservoir hosts to humans via contaminated soil and water. Given the transmission route, our findings of *Leptospira* circulating in environmental reservoirs suggest a risk for environmentally transmitted disease. The high presence of *Leptospira* DNA (10.2% in water and 23.1% in soil) and the isolation of *Leptospira* spp. from environmental samples (6.5% from water and 30.8% from the soil) around the training areas confirm the distribution of *Leptospira* and increase the likelihood of leptospirosis transmission to humans. Most positive *Leptospira* DNA belonged to an intermediate clade, which is also associated with pathogenic strains. One water sample tested positive for *L. interrogans* DNA, but we were unable to isolate the bacteria in culture, possibly due to varying *L. interrogans* decay rates depending on environmental conditions [39]. In contrast, the causative agents of leptospirosis can persist in the soil, water, or sewage for extended periods, and they can even multiply in waterlogged soil, particularly after flooding [39,40]. The pH and temperature ranges observed in this study suggest conditions favorable for *Leptospira*’s survival. To address this limitation and gain a more comprehensive understanding of environmental factors influencing *Leptospira* persistence and transmission, we recommend incorporating soil composition analysis in future studies. However, the infective strains of intermediate *Leptospira* have not been widely found in Thailand, with the exception of one *L. wolffii* isolate from a suspected human case [41]. The increasing recognition of intermediate clades causing human leptospirosis is concerning, especially given the naïve personnel deployed for training in Thailand. Two provinces, Chantaburi and Rayong, were consistently selected as ENCAP project sites over several years and were found to have high *Leptospira* carriage rates in rodents and persistent environmental contamination. This suggests a significant risk of leptospirosis from rodent populations and environmental reservoirs around the exercise sites where military personnel are stationed.

Researchers from Thailand’s MoPH found that wearing long pants significantly reduced the risk of leptospirosis, while having more wounds increased the risk, even after adjusting for other factors [42]. This aligns with the established link between leptospirosis and exposure in agricultural [10] and outdoor environments [43], which particularly affects agricultural workers and local communities, imposing a substantial health and economic burden [44]. The majority of leptospirosis patients reported by MoPH are agricultural workers, especially those involved in rice cultivation, creating environments where rodents, livestock, and humans interact, facilitating bacterial transmission. Infected animals shed *Leptospira* through urine, contaminating the environment, especially during the rainy season when flooding spreads the bacteria. However, a limitation of this study is its exclusive focus on rodents, neglecting other potential reservoirs, such as livestock, dogs, and cats, which may also contribute to environmental contamination and human exposure. The lack of surveillance of these additional reservoir hosts, particularly cattle [45], dogs [46], and cats [47], leaves their role in bacterial transmission unaddressed. This is particularly concerning in areas with dairy farming, where the risk of *Leptospira* transmission is known to be higher [35]. Furthermore, the study’s focus on military training sites limits its geographic scope, highlighting the need for broader surveillance in diverse locations and the inclusion of multiple reservoir hosts to better understand the dynamics of *Leptospira* transmission and identify critical control points for prevention.

A recent report highlights the high risk of infectious diseases, particularly bacterial infections, for US service members traveling outside of the continental United States. Environmental exposures during daily activities, such as wading in fresh water, encountering rodents, working in abandoned buildings, or visiting animal slaughter areas, are key factors in the transmission of *Leptospira* [48]. Documented leptospirosis outbreaks among deployed service members further underscore this risk, including cases among US Marine trainees in Okinawa, Japan [16,49], US military personnel in Honduras [50], and US troops in Guam [15], all associated with environmental exposure during training or deployments. Given the absence of a human vaccine, prophylaxis is a practical disease control strategy for military personnel who cannot consistently avoid exposure. Evidence from a 2014 leptospirosis outbreak among US Marine trainees in Okinawa, Japan highlighted the influence of environmental factors on disease incidence, suggesting that the current prophylactic regimen may be ineffective in high-risk environments and that reevaluation of doxycycline prophylaxis could be indicated [16]. In support of these findings, a systematic review by Guzman et al. (2021) [18] evaluated the effectiveness of antibiotics for treating leptospirosis and explored chemoprophylaxis strategies, revealing that a single 200 mg dose of doxycycline taken before exposure to high-risk environments significantly reduced new symptomatic cases of leptospirosis. Doxycycline’s effectiveness was also confirmed in residents exposed to flooding in southern Thailand [51].

Leptospirosis is often underreported in endemic regions. This underreporting leads to an underestimation of risks for personnel participating in exercises in countries like Thailand. Military personnel face added risk because leptospirosis symptoms frequently overlap with those of other tropical diseases, such as malaria and dengue. This overlap can lead to misdiagnosis and severe outcomes, including death from Weil’s disease, a serious complication that often involves kidney failure. Given the importance of surveillance in assessing such risks, WRAIR-AFRIMS implemented a pre-exercise surveillance program to better understand the incidence of leptospirosis in Thailand. US military personnel are particularly vulnerable due to immune naïveté and increased exposure to *Leptospira* due to activities during deployment. In contrast, individuals living in endemic regions commonly encounter *Leptospira* during daily activities, which can lead to the development of naturally acquired immunity against severe disease. Addressing the underestimated burden of leptospirosis requires comprehensive surveillance in endemic areas, combined with historical outbreak data and relevant research. These efforts provide valuable strategies to mitigate the impact of infectious diseases on military personnel and improve readiness, prevention, and treatment initiatives. Additionally, the underreporting of mild or asymptomatic human cases limits understanding of the true disease burden, emphasizing the need for community-based or syndromic surveillance to complement existing efforts. Such enhanced surveillance can support the Thai MoPH in controlling disease incidence and strengthening the country’s surveillance system. Our study highlights the importance of understanding *Leptospira*’s prevalence in primary hosts and environmental reservoirs to better assess the risk of leptospirosis. Future research will focus on exploring the diversity of pathogenic *Leptospira* strains to evaluate clonal complexes across Thailand, further contributing to improved disease control and prevention strategies.

Our study provides key insights into leptospirosis in military exercise areas in Thailand, aiding Force Health Protection decisions regarding emerging infectious disease threats to military units. By integrating pre-exercise surveillance data with historical outbreak information and relevant research, we improve our capacity to mitigate the impact of infectious diseases on military readiness. These findings also support the Thai MoPH in controlling the disease and enhancing surveillance in areas where it is often underreported. Molecular prevalence methods are crucial for detecting chronic, asymptomatic leptospirosis in small mammals and environmental reservoirs. These techniques underscore the transmission risks of *Leptospira* shed in carriers’ urine and contaminated environments, reinforcing the importance of an approach that connects human, animal, and environmental health.

## Figures and Tables

**Figure 1 microorganisms-13-00029-f001:**
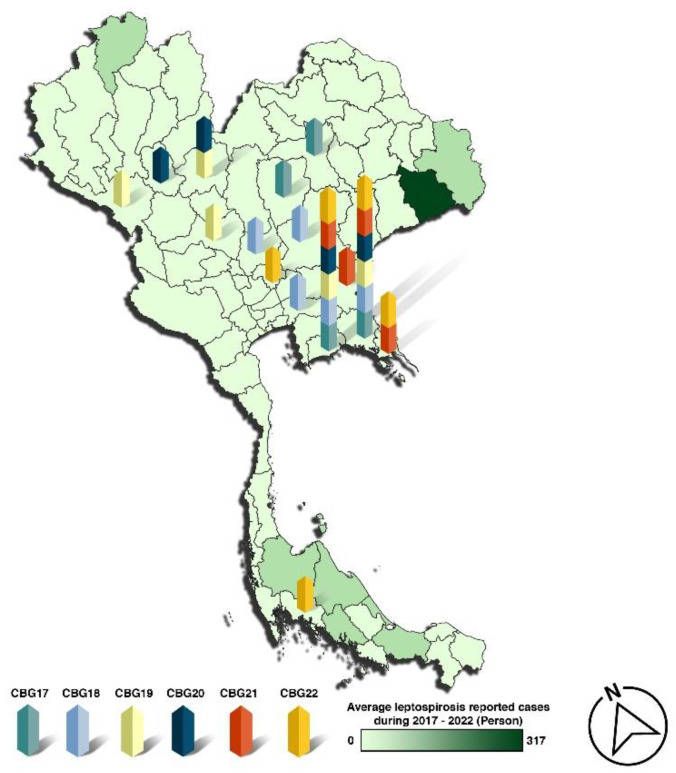
Surveillance locations of CBG training sites (2017–2022) in Thailand and reported passive surveillance leptospirosis cases from the Thai MoPH. Surveillance sites for each year are represented by the code “CBG” and distinguished by color.

**Figure 2 microorganisms-13-00029-f002:**
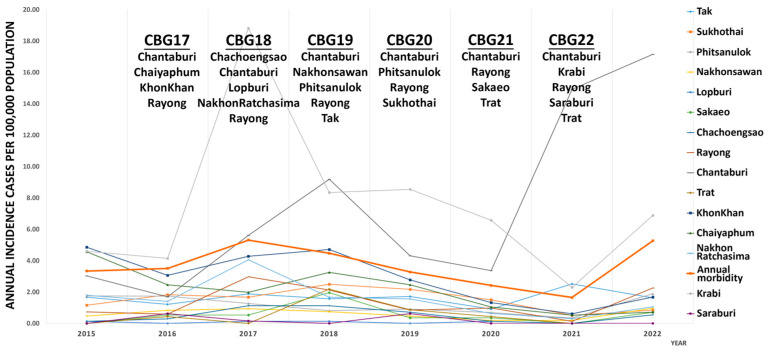
Annual leptospirosis incidence per 100,000 population in selected provinces, aligned with training site locations and the national average morbidity rate in Thailand from 2015 to 2022. The CBG training provinces for each year were listed under the respective training year (CBGFY). The annual incidence per 100,000 population and morbidity rates in selected provinces from 2015 to 2022 were obtained from the MoPH Annual Epidemiology Surveillance Report [11].

**Figure 3 microorganisms-13-00029-f003:**
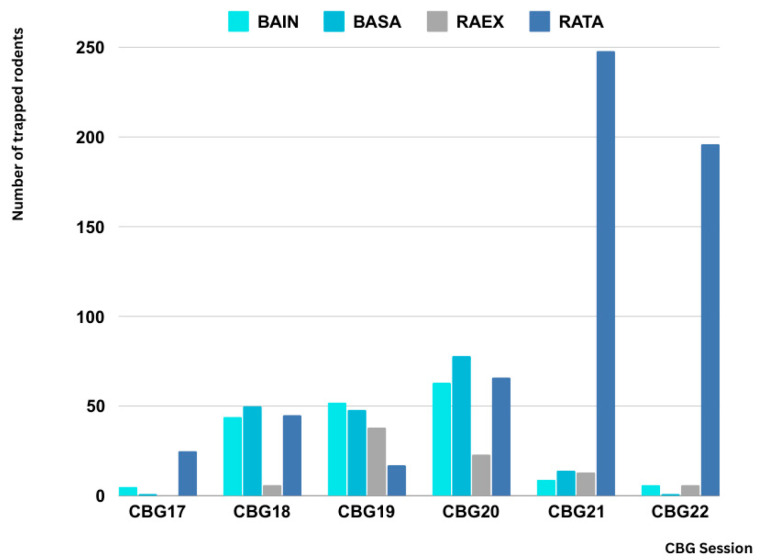
Number of trapped rodents by species across trapping sessions (2017–2022) in CBG sites. The graph shows the four most common rodent species captured during each trapping session from CBG17 to CBG22. Species include *B. indica* (BAIN), *B. savilei* (BASA), *R. exulans* (RAEX), and *R. rattus* (RATA), represented by different shades. The number of rodents trapped varies by session and species, with *R. rattus* showing a significant increase in CBG21 and CBG22.

**Figure 4 microorganisms-13-00029-f004:**
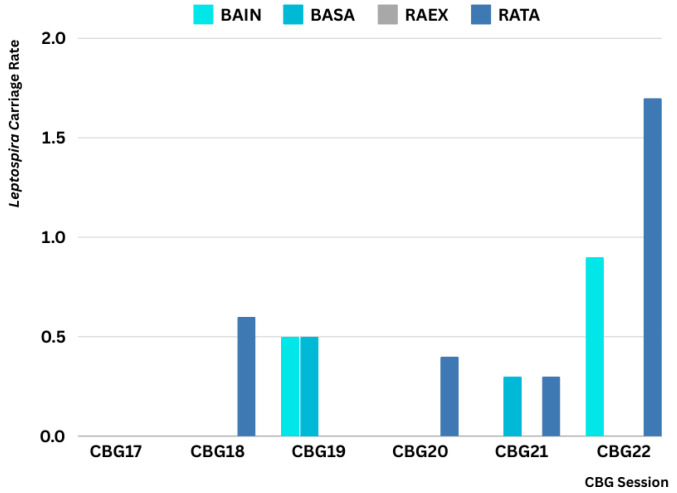
*Leptospira* carriage rate by rodent species across trapping sessions (2017–2022) in CBG sites. The graph depicts the rate of *Leptospira* carriage in the four most commonly trapped rodent species, including *B.*
*indica* (BAIN), *B.*
*savilei* (BASA), *R.*
*exulans* (RAEX), and *R.*
*rattus* (RATA), across sessions from CBG17 to CBG22. While the data reveal fluctuations in carriage rates among the species and sessions, with *R. rattus* exhibiting the highest rate in CBG22, *R. exulans* showed no detectable *Leptospira* carriage throughout the study period.

**Figure 5 microorganisms-13-00029-f005:**
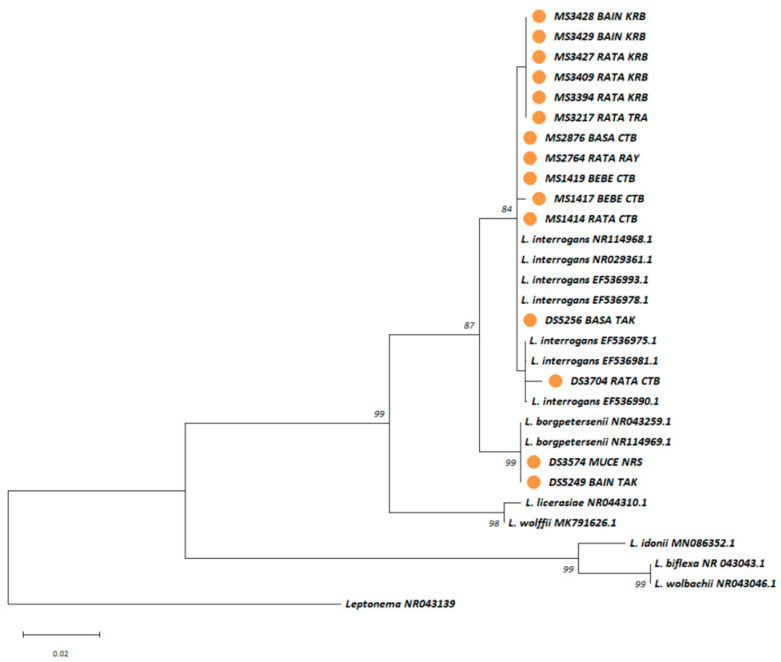
A maximum likelihood phylogenetic tree was constructed based on partial 16S rRNA gene sequences (443 base pairs) of *Leptospira* using the Kimura 2-parameter nucleotide substitution model. Sequences from the 15 *Leptospira*-positive rodents are represented by orange circles. Bootstrap estimates (1000 replicates) are shown above the branches for nodes with >50% support. The abbreviations used in the figure for rodent species and provinces are as follows: BAIN = *Bandicota indica*, BASA = *Bandicota savilei*, BEBE = *Berylmys berdmorei*, MUCE = *Mus cervicolor*, RATA = *Rattus rattus*, CTB = Chantaburi, KRB = Krabi, NRS = NakhonRatchasima, RAY = Rayong, TAK = Tak, and TRA = Trat.

**Figure 6 microorganisms-13-00029-f006:**
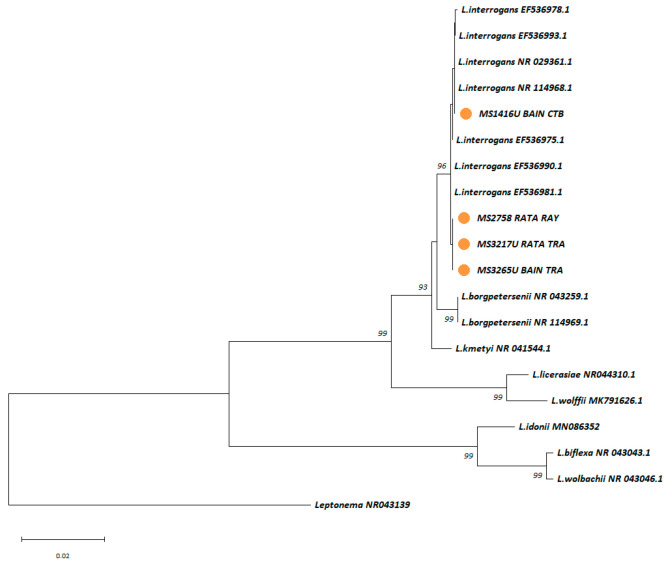
A maximum likelihood phylogenetic tree was constructed based on nearly full-length 16S rRNA gene sequences using the Kimura 2-parameter nucleotide substitution model. Sequences from the four *Leptospira* isolates obtained from trapped rodents are represented by orange circles. Bootstrap estimates (1000 replicates) are shown above the branches for nodes with >50% support. The abbreviations used in the figure for rodent species and provinces are as follows: BAIN = *Bandicota indica*, RATA = *Rattus rattus*, CTB = Chantaburi, RAY = Rayong, and TRA = Trat.

**Figure 7 microorganisms-13-00029-f007:**
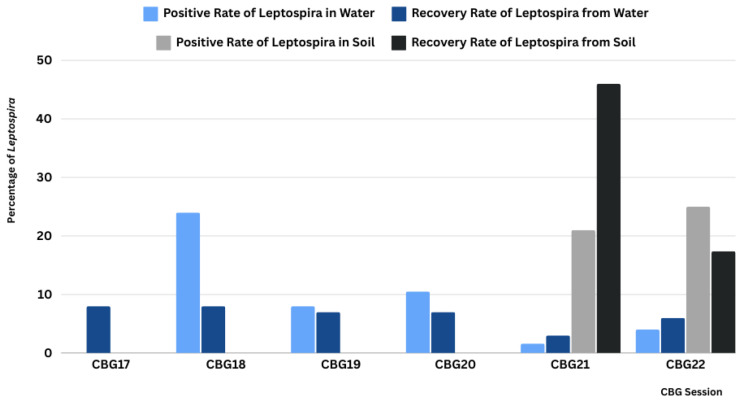
*Leptospira*’s persistence and natural circulating *Leptospira* in environmental sources around exercise sites in 15 provinces over the surveillance years.

**Figure 8 microorganisms-13-00029-f008:**
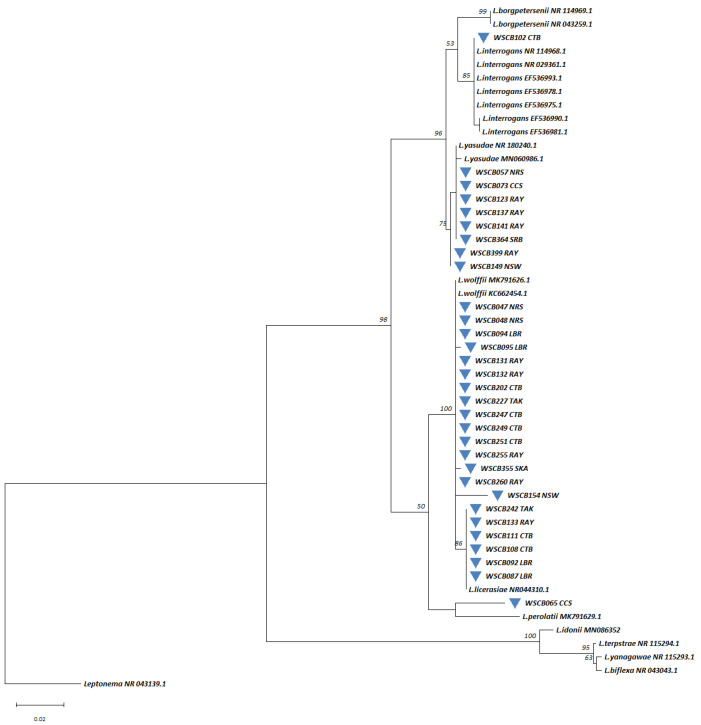
A maximum likelihood phylogenetic tree was constructed based on partial 16S rRNA gene sequences (443 base pairs) of *Leptospira* using the Kimura 2-parameter nucleotide substitution model. The phylogenetic relationships and geographical distribution of the 31 *Leptospira*-positive water samples are represented by blue triangles. Bootstrap estimates (1000 replicates) are shown above the branches for nodes with >50% support. Abbreviations for the provinces are as follows: CCS = Chachoengsao, CTB = Chantaburi, LBR = Lopburi, NRS = NakhonRatchasima, NSW = Nakhonsawan, RAY = Rayong, SKA = Sakaeo, SRB = Saraburi, and TAK = Tak.

**Figure 9 microorganisms-13-00029-f009:**
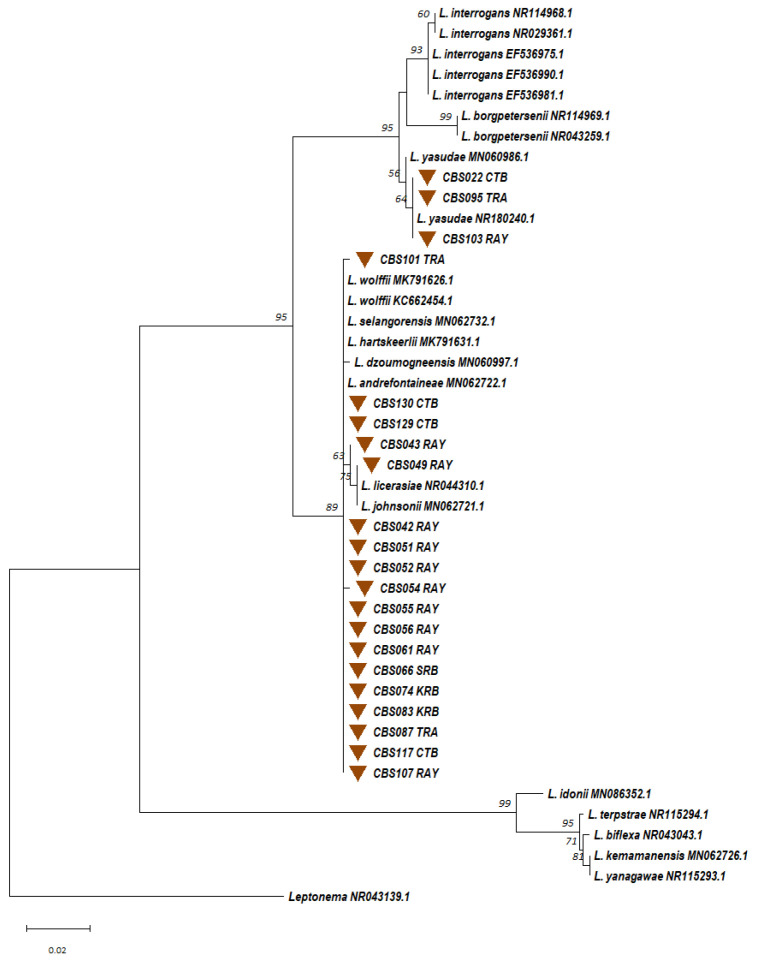
A maximum likelihood phylogenetic tree was constructed based on partial 16S rRNA gene sequences (443 base pairs) of *Leptospira* from soil samples using the Kimura 2-parameter nucleotide substitution model. The phylogenetic relationships and geographical distribution of the 21 *Leptospira*-positive soil samples are represented by brown triangles. Bootstrap estimates (1000 replicates) are shown above the branches for nodes with >50% support. Abbreviations for the provinces are as follows: CTB = Chantaburi, KRB = Krabi, NRS = Nakhon Ratchasima, RAY = Rayong, SRB = Saraburi, TAK = Tak, and TRA = Trat.

**Figure 10 microorganisms-13-00029-f010:**
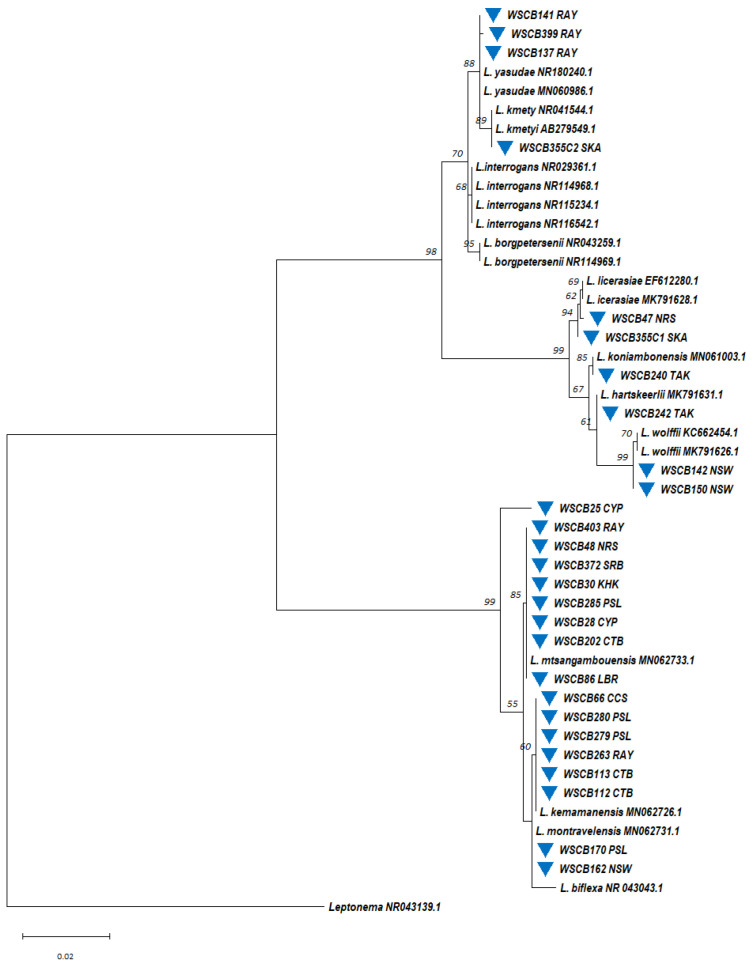
A maximum likelihood phylogenetic tree was constructed based on nearly full-length 16S rRNA gene sequences of *Leptospira* from water isolates using the Kimura 2-parameter nucleotide substitution model. The phylogenetic relationships and geographical distribution of the 27 *Leptospira* water isolates are represented by blue triangles. Bootstrap estimates (1000 replicates) are shown above the branches for nodes with >50% support. Abbreviations for the provinces are as follows: CCS = Chachoengsao, CYP = Chaiyaphum, CTB = Chantaburi, KHK = Khonkhan, LBR = Lopburi, NRS = NakhonRatchasima, NSW = Nakhonsawan, PSL = Phitsanulok, RAY = Rayong, SKA = Sakaeo, SRB = Saraburi, and TAK = Tak.

**Figure 11 microorganisms-13-00029-f011:**
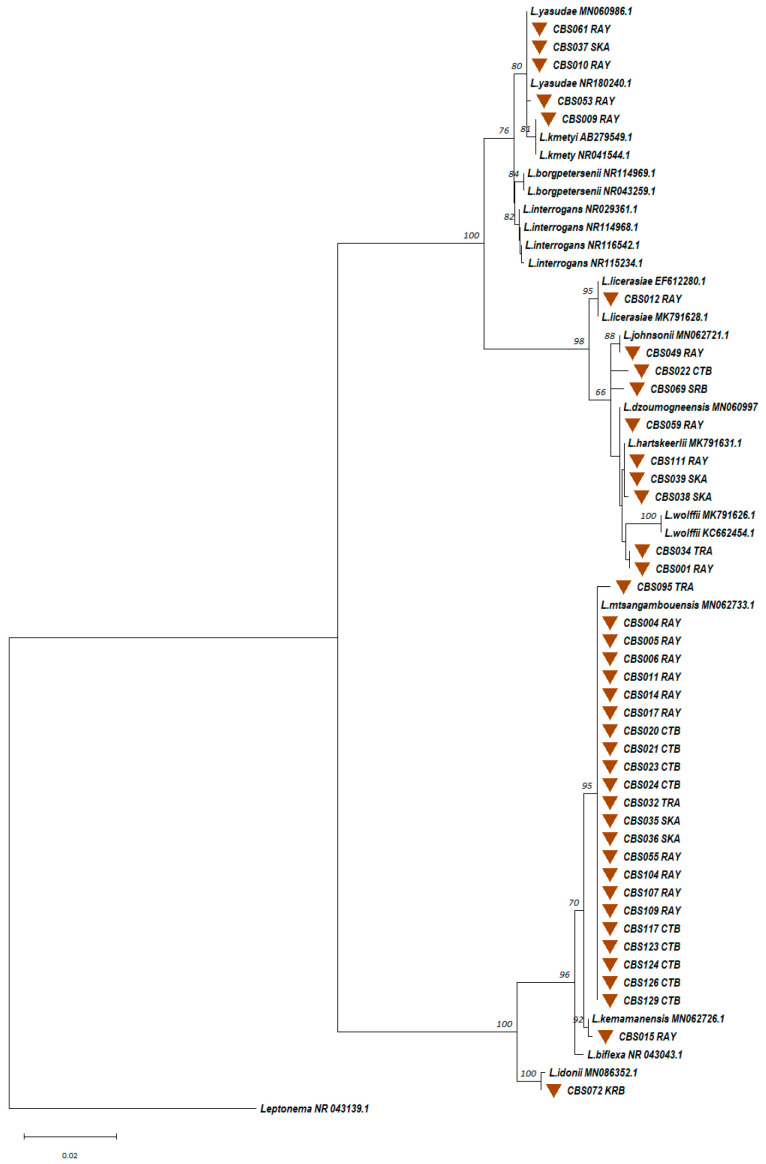
A maximum likelihood phylogenetic tree was constructed based on nearly full-length 16S rRNA gene sequences of *Leptospira* from soil isolates using the Kimura 2-parameter nucleotide substitution model. The phylogenetic relationships and geographical distribution of the 40 soil *Leptospira* isolates are represented by brown triangles. Bootstrap estimates (1000 replicates) are shown above the branches for nodes with >50% support. Abbreviations for provinces are as follows: CTB = Chantaburi, KRB = Krabi, RAY = Rayong, SKA = SaKaeo, SRB = Saraburi, and TRA = Trat.

## Data Availability

All original data supporting the findings of this study are included in the article and the Appendix A. Further inquiries regarding the data can be directed to the corresponding author.

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
