# Peer review of "Leptospirosis Risk Assessment in Rodent Populations and Environmental Reservoirs in Humanitarian Aid Settings in Thailand"

_microorganisms, 2024, doi:10.3390/microorganisms13010029_

Round 1
Reviewer 1 Report
Comments and Suggestions for Authors
The manuscript by Panadda Krairojananan titled "Leptospirosis Risk Assessment in Rodent Populations and Environmental Reservoirs in Humanitarian Aid Settings in Thailand" devoted to the study of Leptospira isolates collected in Thailand from rodents and the environment. The authors performed 16S rRNA gene sequencing and MLST of the isolates. The data have been deposited in the GenBank database and the MLST database. The manuscript contains extensive material, including 10 tables, all of which are included in the supplementary materials, and 11 figures. However, there are still some issues to address regarding style and the English language. For example, in line 269, the phrase "distributed geographically distributed" is confusing. Additionally, in the sentences in lines 38-39, the phrase ", so" appears twice in a row.
Other issues are as follows:
First of all, the Latin terms "Leptospira", "L. interrogans", etc. need to be italicized throughout the manuscript starting from line 101.
Moreover, gene names need to be italicized as well (line 130 "rrs", and lines 146, 152, etc.).
Line 153: "TM" should be in uppercase.
Line 155: Which specific database in GenBank was compared? There are several.
Line 157: Table numbering should follow the order of their mention in the text. The same applies to the figures. Figures should appear after their mention in the text.
Line 160: Remove "(http://www.megasoftware.net/)"
Line 181: Remove "(IBM SPSS Statistics for Windows, Version 26.0."
Figure 2: Make the labels larger; they are illegible.
Are Figures 3 and 4 identical? I do not see any difference, and moreover, Figure 4 is not mentioned in the text.
Comments on the Quality of English LanguageI would recommend English editing.
Reviewer 2 Report
Comments and Suggestions for Authors
The paper's strengths lie in its rigorous multi-year surveillance program, which utilizes advanced molecular techniques like real-time PCR, dark-field microscopy, and 16S rRNA gene sequencing to ensure accurate detection of Leptospira in rodent populations and environmental reservoirs. These findings offer valuable insights into Leptospira transmission dynamics in military and humanitarian aid settings, aiding Force Health Protection strategies and informing public health interventions in Thailand​.
However, the following items should be considered to improve the paper:
1. The study focuses solely on rodents, neglecting other potential reservoirs like livestock, dogs, and cats, which could contribute to environmental contamination and human exposure.
2. Conducting sampling only during the dry season may underestimate Leptospira prevalence, as transmission rates increase during the rainy season.
3. The underreporting of mild or asymptomatic human cases limits understanding of the true disease burden, highlighting the need for community-based or syndromic surveillance.
4. Lack of detailed environmental data, such as temperature, pH, and soil composition, reduces insight into factors influencing Leptospira persistence and transmission.
5. Incomplete molecular typing of all positive samples restricts the understanding of the full genetic diversity of circulating Leptospira strains.
6. Limited geographic coverage focused on military training sites may not reflect broader regional trends, underscoring the need for expanded surveillance locations.
